# Selective plasticity of fast and slow excitatory synapses on somatostatin interneurons in adult visual cortex

Bryce D. Grier[1,2,5], Samuel Parkins[2,3], Jarra Omar[2] & Hey-Kyoung Lee ●[1,2,3,4] ✉

Somatostatin-positive (SOM) interneurons are integral for shaping cortical processing and their dynamic recruitment is likely necessary for adaptation to sensory experience and contextual information. We found that excitatory synapses on SOMs in layer 2/3 (L2/3) of primary visual cortex (V1) of mice can be categorized into fast (F)- and slow (S)-Types based on the kinetics of the AMPA receptor-mediated current. Each SOM contains both types of synapses in varying proportions. The majority of local pyramidal neurons (PCs) make unitary connections with SOMs using both types, followed by those utilizing only S-Type, and a minority with only F-Type. Sensory experience differentially regulates synapses on SOMs, such that local F-Type synapses change with visual deprivation and S-Type synapses undergo plasticity with crossmodal auditory deprivation. Our results demonstrate that the two types of excitatory synapses add richness to the SOM circuit recruitment and undergo selective plasticity enabling dynamic adaptation of the adult V1.

Cortical function and plasticity are intricately regulated by coordinated actions of inhibitory interneurons. Somatostatin-positive interneurons (SOMs) constitute one of the 3 major groups of inhibitory interneurons present in the cortex[1]. Among the interneuron types, parvalbumin-positive interneurons (PVs) are the most well-characterized. PVs participate in fast feedforward and feedback inhibition that targets the soma and proximal dendrites of excitatory neurons[2], increases spike reliability, and reduces signal-to-noise ratio[3], as well as provide gain control[4,5] and feature selectivity[6]. In contrast, SOMs provide localized feedback inhibition to the dendrites of excitatory neurons present in the cortex by integrating cortical activity, predominantly at higher activity levels[1], and respond to long-range contextual information[7]. In the primary visual cortex (V1), SOMs participate in visual processing by supporting surround suppression[8] and feedback receptive fields[9], in addition to helping establish binocular matching[10] in layer 2/3 (L2/3) pyramidal neurons. The degree to which SOMs respond to visual stimuli is also heavily dependent on behavioral state[11]. Collectively, these results suggest that SOMs in V1 L2/3 are

integral to visual processing and that dynamic recruitment of the SOM circuit is expected to facilitate context-dependent adaptation of V1 processing.

While there is a good understanding of the inhibition mediated by SOMs and the role that SOMs play in cortical circuits, excitatory synapses in SOMs are rather understudied[7,12]. To date, studies of excitatory synapses on SOMs have largely been carried out in the juvenile brain. In juveniles, the single-spike failure rate at glutamatergic synapses onto SOMs is as high as 60–80%[7,13]. This low release probability endows excitatory synapses on SOMs with frequency-dependent short-term facilitation[13–15]. It is unknown, however, whether these properties persist in adult SOMs. It has been reported that in L2/3 PVs, local excitatory inputs become less depressing with development[16], and this suggests the possibility that the dynamics of excitatory synapses on SOMs might also be developmentally regulated. Recent studies highlight that a rapid and transient decrease in excitatory inputs to PVs is critical for producing transient disinhibition necessary for inducing V1 plasticity in juveniles[17–21]. Whether excitatory

[1]Solomon H. Snyder Department of Neuroscience, Johns Hopkins School of Medicine, Baltimore, MD 21205, USA. [2]Zanvyl-Krieger Mind/Brain Institute, Johns Hopkins University, Baltimore, MD 21218, USA. [3]Cell Molecular Developmental Biology and Biophysics (CMDB) Graduate Program, Johns Hopkins University, Baltimore, MD 21218, USA. [4]Kavli Neuroscience Discovery Institute, Johns Hopkins University, Baltimore, MD 21218, USA. [5]Present address: Bionic Sight, New York, NY 10022, USA. ✉e-mail: heykyounglee@jhu.edu

synapses on SOMs undergo plasticity with changes in sensory experience is unknown, especially in the adult cortex.

Here we show that SOMs in V1 L2/3 of adult mice possess two distinct types of excitatory synapses, fast- (F) and slow- (S) Types, as identified by the kinetics of the AMPA receptor-mediated synaptic currents. The two types were evident when recording miniature excitatory postsynaptic currents (mEPSCs), and were also observed in unitary excitatory currents (uEPSCs) recorded from pairs of a presynaptic pyramidal cell (PC) and a postsynaptic SOM. The majority of connected PC to SOM pairs displayed both F- and S-Type uEPSCs, suggesting that a single presynaptic neuron can make functionally distinct synapses. In addition, we found pairs that were solely S-Type or solely F-Type. All 3 types of paired connections exhibit short-term facilitation, consistent with that reported in juveniles[13–15], but they differ in their stimulus frequency dependence and how they are regulated by visual deprivation. Similar to what is known in juveniles[22], SOMs in adult V1 L2/3 display stimulus-dependent asynchronous release, which was modified by visual deprivation only in the S-type synapses. Visual deprivation generally dampened activity propagation from local PC to SOMs, except for high frequency transmission at F-Type connections. In contrast, auditory deprivation selectively increased the frequency of S-Type mEPSCs. Our results suggest that SOM interneurons in adult V1 L2/3 display two distinct types of excitatory synapses, which are regulated independently, and in distinct manners, by different changes in sensory experience to display a rich repertoire of experience-dependent plasticity.

## Results

### Two types of excitatory synapses on V1 L2/3 SOMs

In order to investigate excitatory synapses on SOMs in adult circuitry, we used the GIN transgenic line[23], in which a subset of SOMs termed Martinotti cells (MCs) express green fluorescence protein (GFP)[24] (Fig. 1a). In L2/3 of primary sensory cortices, SOMs are MCs[12,24] and categorized as Sst-MET-3 based on the Morphology-Electrophysiology-Transcriptomics (MET) identification method[25]. We began by recording mEPSCs from adult V1 L2/3 SOMs in mice with normal visual experience (normal reared; NR). When we examined the kinetics of the mEPSCs recorded from SOMs, we noticed a marked heterogeneity. Within single cells, mEPSC kinetics varied greatly, giving the appearance of distinct populations of mEPSCs (Fig. 1b). Through plotting mEPSC charge transfer against amplitude, we observed a relationship between these two parameters that was suggestive of two distinct populations of mEPSCs that could be delineated by their kinetics (Fig. 1c–f). This relationship was visible in every NR cell recorded (Fig. S1). There was, however, variability in the relationship between mEPSC amplitude and charge transfer across cells with variable slopes of the two "arms" leading to smaller or larger gaps between the two putative groups (Fig. S1). To determine whether the observed heterogeneity in kinetics was due to dendritic filtering, we pooled mEPSCs across cells and compared mEPSC amplitude to 10–90% rise time and found no negative correlation between these values (Fig. S2). This suggests that the observed events with slower kinetics are not simply reflecting dendritic filtering of distal synapses[26].

To separate the two putative populations of mEPSCs, we first employed a novel cost function to rotate the mEPSC amplitude and charge data for each cell in an unbiased fashion (see Methods). This rotation allowed us to effectively fit a one-dimensional two-component Gaussian mixture model to each cell and pull out two clusters of mEPSCs in an unbiased manner (Fig. 1b, c). We termed these two groups S-Type (slow) and F-Type (fast), and the distinction between the two is clear even when mEPSCs are pooled across a heterogeneous population of cells (Fig. 1e, f). When viewed across the population of recorded cells, S-Type events are more prevalent (Fig. 1g), but individual cells display a high variability in their mEPSC composition, with the proportion in each cell varying across a continuum from S-Type

dominant to F-Type dominant (Fig. 1g; Fig. S1). The variability across cells in the slope of the within-group charge to amplitude relationship leads to some overlap between the two groups when mEPSCs are pooled across cells and plotted (Fig. 1e). When comparing the two groups we found that S-Type events had a significantly greater charge transfer whereas F-Type events had a significantly greater amplitude, in addition to differences in the kinetics (Fig. S3). These observed differences in amplitude and charge transfer suggests that different rules govern how well the two types of mEPSCs depolarize the postsynaptic membrane and summate to bring a cell to a threshold. The difference in charge transfer further supports that S-Type events are not merely filtered F-Type events. With dendritic filtering, as the amplitude of synaptic current diminishes with distance from the soma, the charge transfer largely remains the same[27]. Thus, while the slower rise and decay kinetics (Fig. S3) of S-Type events suggests that they may be more distally located[28], they appear to be a population of mEPSCs that is distinct from F-Type events.

To rule out the possibility that we may be recording an additional ionotropic current besides AMPAR-mediated mEPSCs, we bath applied an AMPA receptor antagonist, NBQX, and confirmed that both types of mEPSCs were indeed blocked (Fig. S4a). Having determined that the currents were indeed glutamatergic, we thought that the fast kinetics observed in some mEPSCs may reflect the presence of $Ca^{2+}$-permeable AMPA receptors (CP-AMPARs)[29]. We found that bath application of 1-naphthyl acetyl spermine (Naspm), a CP-AMPAR blocker, reduced the amplitude of mEPSCs (Fig. S4b), supporting the idea that CP-AMPARs are present at excitatory synapses on L2/3 SOMs. This stands in contrast to layer 5 (L5) SOMs in juveniles, which have been shown to lack CP-AMPARs[30]. When S- and F-Type mEPSCs were analyzed separately, we found that the CP-AMPAR blocker selectively reduced the amplitude of S-Type events (Fig. S4c). This suggests the existence of CP-AMPARs with unusual slow kinetics, which are likely associated with either the γ−4 or γ−8 isoforms of the transmembrane AMPA receptor regulatory protein (TARP)[31]. In support of this interpretation, prior transcriptomics analysis shows expression of γ−4, in addition to other TARPs, in V1 L2/3 SOMs[32].

We also determined that the presence of two types of mEPSCs is not unique to adult SOMs. We observed the same S- and F-Type mEPSCs in SOMs of juvenile mice (P14-P16), in proportions similar to those seen in adult SOMs (Fig. S5). The mEPSC frequency in juveniles, however, was significantly lower for both S- and F-Types than in adults (Fig. S5a). The average amplitude was also lower as compared to adults, with F-Type events reaching statistical significance (Fig. S5a). These results suggest that there is developmental maturation of excitatory synapses, but that the proportion of S- and F-Type mEPSCs is not subject to developmental regulation (Fig. S5b, c).

### Local excitatory connections utilize S- and/or F-Types synapses

From our initial observations, it was unclear whether the two mEPSC types represent distinct inputs. To investigate this, we performed paired whole-cell recordings between V1 L2/3 PCs and SOMs and elicited trains of action potentials in the presynaptic PC (Fig. 2a). Through recording both local PC-driven unitary EPSCs (uEPSCs) and spontaneous EPSCs (sEPSCs) in the same SOM cell we were able to classify unitary events as either S-Type or F-Type in an unbiased manner (Fig. 2b, c). This was accomplished by clustering sEPSCs on a cell-by-cell basis as we did with mEPSCs, rotating the uEPSC data by the same degree as the sEPSC data and then, classifying the uEPSCs using the Gaussian mixture model that was fit to the sEPSCs. This allowed us to assess the uEPSCs without assuming a priori that there would be two types of events within a given set of uEPSCs. We observed that the uEPSCs evoked in some pairs displayed homogeneous kinetics (Fig. 2c). When this was the case, the pairs were labeled as either S-Type or F-Type pairs, depending on whether the majority of group membership posterior probabilities were clustered near 0 or 1, which

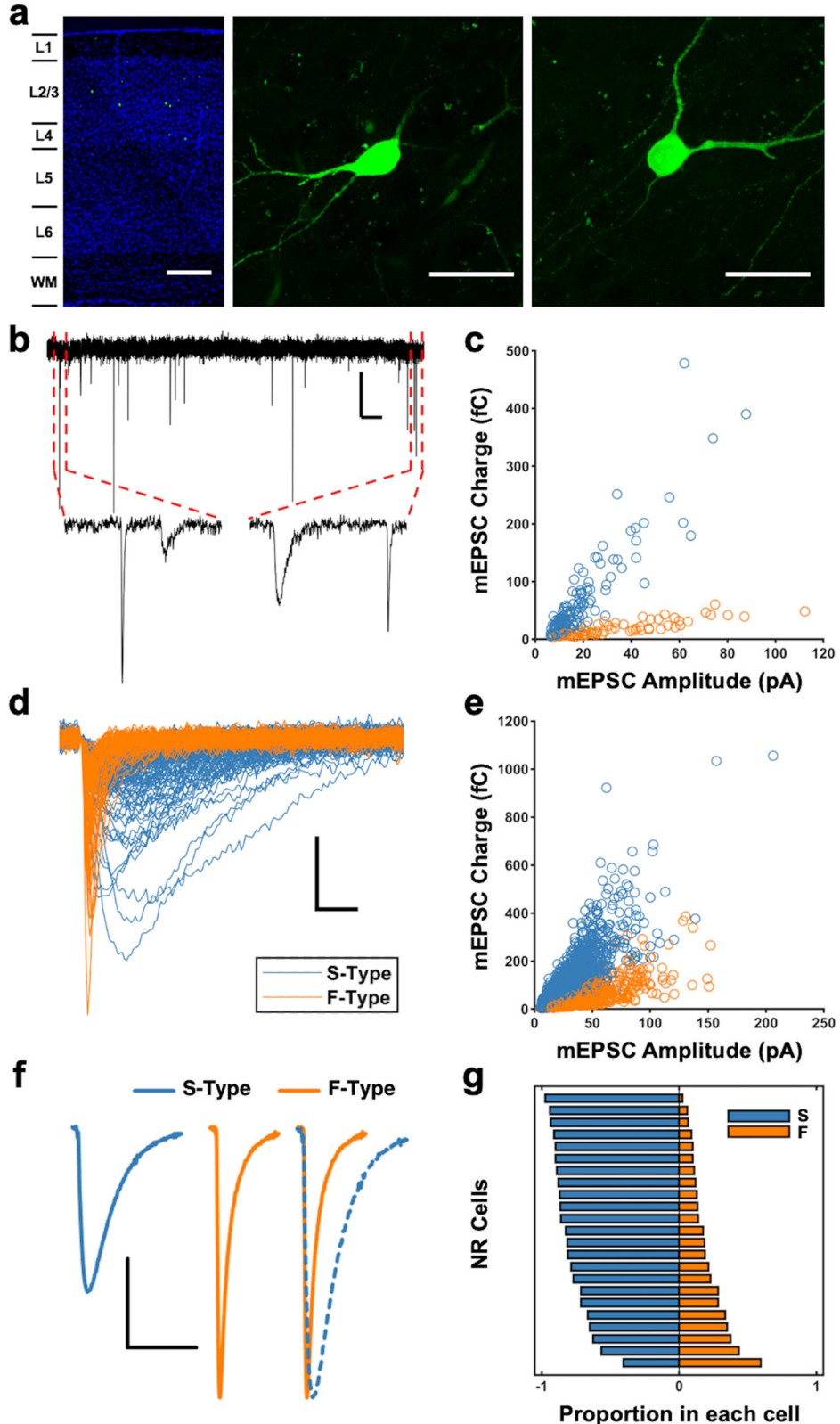

indicated a high probability of belonging to F-Type or S-Type, respectively. We also observed, however, that in some pairs the uEPSCs displayed heterogeneous kinetics akin to those in the sEPSCs (Fig. 2c). Within these pairs, the majority of group membership posterior probabilities were close to both 0 and 1, indicating high probabilities that some uEPSCs were F-Type and others were S-Type. Together, these observations suggested the presence of both types of EPSCs in

these pairs, which we termed Mixed pairs. Thus, it is clear that two types of EPSCs can be observed at the levels of both quantal release and evoked release. Across the population of recorded pairs, we found Mixed to be the most prevalent, followed by S-Type only, and then F-Type only (Fig. 2b). This finding is novel, as previous reports of kinetically distinct excitatory currents in a given cell found that they were contributed by different presynaptic partners (e.g. mossy fiber

**Fig. 1 | Two types of mEPSCs in V1 L2/3 SOMs. a** Confocal images of V1 sections from SOM-GFP mice. Left: A low magnification (10x) tiled image showing sparse GFP labeled cells (green) in V1. Counterstained with DAPI (blue). Scale bar: 200 μm. Middle and Right: Higher magnification (63x) z-stack images of SOM-GFP neurons in V1 L2/3. Scale bars: 30 μm. Similar results are seen in confocal images of brain sections from 5 different SOM-GFP mice and in all of the slices used for whole-cell recordings. **b** Example voltage clamp recording from a SOM cell. SOM mEPSCs display heterogeneous kinetics. Scale bar: 20 pA, 500 ms. Dotted red lines indicate the expanded portion of the trace. **c** Two distinct populations of mEPSCs are visible when plotting the charge versus amplitude of 200 individual mEPSCs from an example cell. Blue: S-Type events. Orange: F-Type events. **d** Overlaid S-Type (blue) and F-Type (orange) mEPSC traces from the example cell in b. Scale bar: 30 pA, 2 ms. **e** Charge and amplitude of mEPSCs pooled together from all the recorded cells in NR group (200 mEPSCs per cell, 23 cells, 10 mice). **f** Average mEPSCs from each synapse type. Left: average S-Type mEPSC. Center: average F-Type mEPSC. Right: Average S- and F-Type mEPSCs scaled to same amplitude (blue: S-Type, orange: F-Type). Scale bar: 10 pA, 10 ms. **g** The proportion of each type of synapse varies across the population of NR SOM cells. Each horizontal bar represents one cell. The position on either side of the center line indicates the relative proportion of each type.

inputs versus CA3 collateral inputs[33]. Our data indicate that for a sizeable population of V1 L2/3 SOMs, the same presynaptic PC makes two types of synapses which are distinguishable by their AMPAR kinetics. Through analyzing multi-cell recordings[34, 35] from the Allen Institute for Brain Science Synaptic Physiology dataset[34, 35] (Fig. S6), we observed single presynaptic PCs synapsing onto two SOM neurons with distinct kinetics (Fig. S6). This suggests that the two types of excitatory synaptic connects can also be segregated according to postsynaptic targets.

## Short-term dynamics differ between S-Type containing and F-Type only local connections

The short-term dynamics of neocortical excitatory synapses are dependent on the identity of the postsynaptic cell[15], and excitatory synapses onto SOMs have been shown to be facilitating in juveniles[13,14]. Given that the short-term synaptic dynamics may change with maturation as reported for PVs[16], we sought to determine whether these synapses were facilitating in adult circuitry and whether there were differences in the dynamics of S-Type, F-Type, and Mixed connections. To study this, during the recordings performed in the paired configuration to determine the pair types (Fig. 2), we elicited trains of 10 presynaptic action potentials at 4 different frequencies (5 Hz, 10 Hz, 20 Hz, 40 Hz) (Fig. 3). Similar to what is reported in juveniles, we found that all 3 types of excitatory connections on V1 L2/3 SOMs are facilitating (Fig. 3b, c).

To analyze these data, we fit linear mixed-effects models to individual connection types and found variations in the exact nature of facilitation across the 3 identified types of pairs (Fig. 3c; Table S1). In both S-Type pairs and Mixed pairs, there was a significant interaction between the effect of stimulus frequency and stimulus number on synaptic strength (AP:Freq in Table S1). In F-Type pairs, however, there was only a significant effect of stimulus number on synaptic strength, such that the degree of facilitation was not affected by the stimulus frequency (AP in Table S1). Additionally, we determined that the observed effects on synaptic strength in S-Type and Mixed pairs were likely due to effects on the postsynaptic potency (Fig. S7a, b; Table S1), whereas in F-Type pairs, the effect on synaptic strength appeared to be driven solely by an effect on success rate (Fig. S7a; Table S1). Together, these data suggest that the properties of short-term facilitation of local excitatory connections on SOMs in adult V1 L2/3 differ between those containing S-Type synapses (S-Type and Mixed pairs) and those predominantly harboring F-Type synapses (F-Type pairs) in that only the former exhibit stimulus frequency dependence. This suggests that S-Type containing inputs more effectively recruits SOMs when there is greater activity.

## Visual deprivation preferentially dampens the short-term dynamics of F-Type local connections

Excitatory inputs from local L2/3 PCs likely allow SOMs to sculpt visual responses of principal neurons to yield properties such as end-stop, surround suppression, and binocular matching[8-10]. This raises the possibility that these inputs may undergo plasticity when visual experience is altered. To test this, we examined whether a week of visual deprivation in the form of dark-exposure (DE) alters the

properties of PC to SOM connections in adult V1 L2/3. We chose this model of visual deprivation, as it is known to produce robust plasticity of the adult V1 circuit by eliciting homeostatic metaplasticity[36, 37] that can reinstate ocular dominance plasticity[38-41] or enable crossmodal sensory adaptation[42-44]. As in NR animals, we recorded uEPSCs from pairs of V1 L2/3 PCs and SOMs and delivered trains of action potentials at several frequencies (Fig. 3c). Following DE, we observed a significant interaction between the effect of stimulus number and stimulus frequency on synaptic strength in both S-Type and Mixed pairs (Fig. 3c, Table S1). In F-Type pairs, however, synaptic strength lost its dependence on stimulus number (Fig. 3c, Table S1). These changes appear to be largely driven by changes in success rate (Fig. S7a, Table S1). Collectively, our data suggest that DE differentially alters the short-term dynamics of these 3 types of local PC inputs to SOMs.

## The strength and short-term dynamics of the three connection types are independently and selectively regulated by experience

By fitting a model to the combined NR and DE pair data, we were able to assess the overall effect of sensory experience on the strength and dynamics of the different types of pairs (Table S2). To begin with, there is a significant effect of experience on overall synaptic strength, with synaptic strength being lower following DE (Fig. 3c, Exp in Table S2). Interestingly there was also a significant effect of experience that varied across the different types of connections (Exp:Type in Table S2), further underscoring the observation that the connection types are independently regulated. We also observed a significant overall interaction between stimulus number and stimulus frequency, and the magnitude of interaction also varied across the 3 types of connections (AP:Freq:Type in Table S2), which further suggests that the 3 different types of synapses have distinct dynamics.

This model also allowed for comparisons of synaptic strength between different connection types, within and across different histories of visual experience through contrasts of estimated marginal means. When comparing within NR pairs, we found that Mixed connections are significantly stronger than both S-Type and F-Type pairs, while there is no overall difference between S- and F-Type pairs (Table S2). In DE pairs, however, there is no difference in overall synaptic strength between any type of connection (Table S2). This surprising result is clarified by comparisons of the same connection type between different visual experience conditions. Here we found that a lack of visual experience resulting from DE yields a selective decrease in the strength of Mixed connections (Fig. 3c, Table S2), thereby normalizing synaptic strength across the types. Thus, it appears that visual experience maintains stronger Mixed connections, suggesting that the presence of distinct types of excitatory connections is necessary for the normal circuit function of V1 L2/3 SOMs.

## Visual deprivation differentially alters the activity transmission based on the connection type

To determine the functional consequence of the plasticity observed at S-Type, F-Type, and Mixed connections from local PCs to SOMs, we played back idealized current traces to V1 L2/3 SOMs from control mice and recorded their spiking output in current clamp (Fig. 4a). These idealized traces were based on the average strength, kinetics,

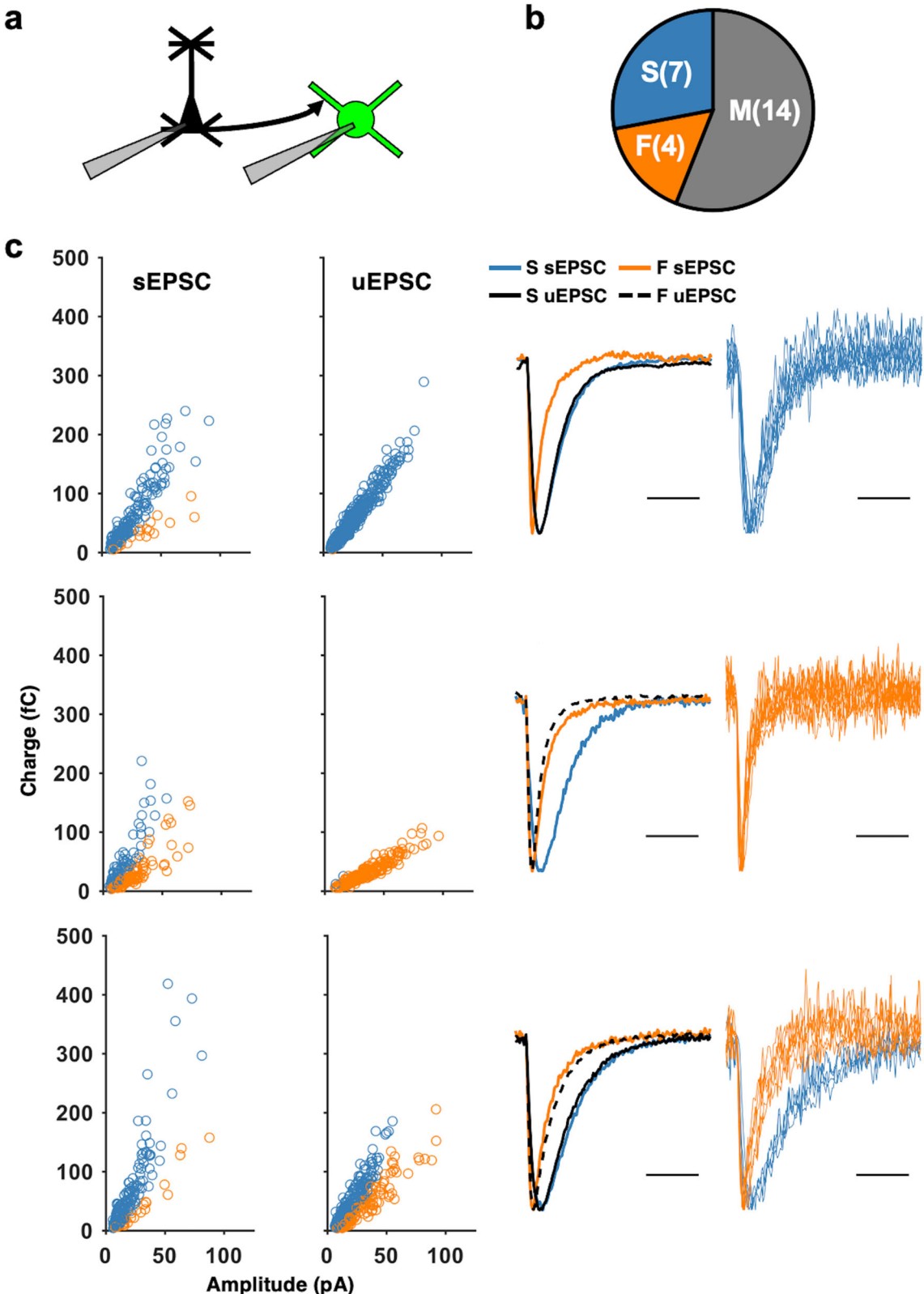

and short-term dynamics of NR and DE uEPSCs of all 3 connection types at 10 and 40 Hz (see Methods). As predicted from the plasticity observed in the strength and short-term dynamics of uEPSCs, we found varied spiking patterns in response to the playback of traces representing different conditions (Fig. 4a, b, and Tables S3 and S4). Overall, our results show significant interactions between the effect of recent sensory experience, synapse type, and the frequency of

playback traces (Type:Exp:Freq in Table S3) on the generation of spikes in postsynaptic SOMs. This suggests that the recruitment of V1 L2/3 SOMs by PCs is highly plastic and subject to shaping by the different synapse types. Further analysis of our results shows a significant effect of connection type (Type in Table S3) with traces representing Mixed connections being the most effective at driving spikes in SOMs across both 10 and 40 Hz synaptic activity (Table S4). There was also a

**Fig. 2 | The majority of local PC to SOM pairs make synapses using both S- and F-Type synapses. a** Schematics of the experiment. Paired whole-cell recordings were made between PCs (black) and SOMs (green) in V1 L2/3. Connectivity was assessed in the PC to SOM direction. **b** The proportion of functionally connected PC to SOM pairs using only S-Type (S, blue), only F-Type (F, orange), or both S- and F-Type (M: Mixed, gray), synapses. The number of pairs is indicated in parentheses. **c** Examples of observed pair types. Left column: Spontaneous EPSCs clustered into S- and F-Types with unitary evoked EPSCs (uEPSCs) from the same SOM overlaid. Middle column: Amplitude-scaled average traces from cell in the corresponding left panel. Blue and orange traces are S-Type and F-Type sEPSCs, respectively. Overlaid solid and dashed black traces are S-Type and F-Type uEPSCs, respectively. Right column: Overlay of 10 randomly selected uEPSC traces from example cells in corresponding left and middle panels. Top row: An example PC to SOM pair with mainly S-Type uEPSCs. Middle row: An example PC to SOM pair with mainly F-Type uEPSCs. Bottom row: An example PC to SOM pair with both S- and F-Type uEPSCs. Note that in all these 3 pair types, sEPSCs cluster into both S-Type (blue) and F-Type (orange). Scale bars: 5 ms.

significant effect of frequency (Freq in Table S3) with trains representing connections that contain S-Type synapses (S-Type and Mixed connections) displaying greater SOM activation at 40 Hz than 10 Hz (Table S4). This frequency dependence was not observed when playing back F-Type uEPSC trains. Together, these results suggest that under normal conditions, local PCs utilizing Mixed connections preferentially drive the recruitment of SOMs and those using S-Types contribute to further recruitment when there is higher activity of local PCs. Additionally, we found a significant effect of recent sensory experience (Exp in Table S3), such that playback of DE traces, in general, produced fewer action potentials in SOMs as compared to the NR traces of the same type and stimulation frequency (Table S3). In lone contrast, however, for 40 Hz F-type traces, the DE trace produced a significantly greater number of action potentials (Table S4). This suggests that visual deprivation overall dampens the propagation of activity from local PCs to SOMs, except at F-type pairs when there is high-frequency presynaptic activity. Idealized traces were chosen over average traces for playback to reduce any confounding effects that natural variability in the sample of recorded traces may have on the spiking output of the stimulated cells. We observed essentially the same results when the average traces from the paired recordings were played back to SOMs (Fig. S8 and Tables S3 and S5).

**Changes in visual experience alter local PC to SOM connectivity**
In addition to driving changes in the short-term dynamics of local excitatory synapses from V1 L2/3 PCs to SOMs, DE resulted in changes in local connectivity. When we measured the intersomatic distance of the pairs of PCs and SOMs (Fig. S9a), there were significant effects of both intersomatic Euclidean distance and DE on the connection probability. Additionally, there was a significant interaction between the effects of intersomatic Euclidean distance and DE. The changes observed with DE show greater connectivity with closer PCs and lower connectivity with distant PCs, suggesting that the connectivity pattern associated with surround-suppression is maintained by visual experience. Finally, we observed no change in the relative proportion of different connection types across the population following DE (Fig. S9b).

**Activity-dependent asynchronous release is observed predominantly at local connections containing S-Type synapses**
It has been shown that asynchronous release occurs at synapses from L5 PCs to L5 MCs in the somatosensory cortex of juvenile rats[22]. We observed evidence of asynchronous release in our recordings of PC to SOM pairs (Fig. 5a, b). We quantified the asynchronous release by counting EPSCs observed after an initial 3-ms window following the peak of each presynaptic action potential (Fig. 5b). In NR adults, we found a significant interaction between the effect of stimulation frequency and stimulus number on the frequency of asynchronous release in S-type and Mixed pairs (Fig. 5c, Table S1). Conversely, F-Type pairs showed comparatively little asynchronous release and no dynamics in the frequency of release. (Fig. 5c, Table S1). Following DE there was a marked change in the dynamics of asynchronous release in S-Type pairs, which led to a lack of facilitation in the frequency of release in these pairs, similar to what was observed in F-Type pairs (Fig. 5c, Table S1).

**Changes in sensory experience drive plasticity in SOM mEPSCs**
To further examine whether there is global adaptation of S- and/or F-Type synapses with changes in sensory experience, we analyzed mEPSCs from V1 L2/3 SOMs following visual deprivation. A week of DE did not alter the amplitude or the frequency of S-Type or F-Type events (Fig. 6a), which suggests that there is no global change in the strength or number of these two types of excitatory synapses on SOM neurons. In addition to participating in shaping visual responses in V1, L2/3 SOMs are thought to participate in a circuit that involves VIP and L1 inhibitory neurons as well as L2/3 PCs, all of which are targets of long-range inputs that convey multisensory and contextual information[42]. Hence, we tested whether crossmodal sensory deprivation might alter excitatory synapses on SOMs. To do this, we performed auditory deprivation (AD) by ototoxic lesioning of the hair cells in the cochleae[43] a week prior to recording mEPSCs in SOMs. Following AD, there was no observed change in the amplitude of either type of mEPSCs (Fig. 6a). There was, however, a significant and selective increase in the frequency of S-Type mEPSCs (Fig. 6a). The increase in frequency did not lead to a significant change in the proportion of S-Type events in AD cells, but there was a significant difference in the variance of proportion across the three types of experience (Fig. 6b, c). Thus, it seems that mEPSCs on adult V1 L2/3 SOMs are regulated not by unimodal changes in activity, but rather, by crossmodal changes in sensory experience. The shift in the composition of synapses across the population of cells following AD suggests plasticity at the level of SOM recruitment that alters the total charge and response integration properties.

## Discussion
Here we report that SOMs in the adult V1 L2/3 express two distinct types of excitatory synapses, S-Type and F-Type, that are distinguishable by the kinetics of the AMPAR-mediated synaptic currents (Fig. 1, Fig. S1). Local V1 L2/3 PCs utilize both S- and F-Type synapses to provide excitatory inputs to SOMs, with about half of the connected pairs utilizing both S- and F-Type synapses, which we term Mixed pairs (Fig. 2). The remaining connections incorporate either S-Type or F-Type synapses, with the majority of these using S-Type. All 3 types of local excitatory connections show short-term facilitation, similar to what is reported in SOMs in juveniles[13, 14], but only the connections containing S-Type synapses exhibit frequency dependence, while the minority population of connections solely utilizing F-Type synapses do not (Fig. 3). This suggests that the recruitment of SOMs by local V1 circuitry is highly versatile in the adult V1 and dependent on the type of synapses. Furthermore, we demonstrate that the specific types of excitatory synapses on SOMs undergo selective plasticity with sensory deprivation in adults (Fig. 6). Visual deprivation altered the short-term dynamics of inputs from local PCs, and the specific effects varied across the connection types (Fig. 3). This visual deprivation-induced plasticity of short-term dynamics also had functional consequences for activity-dependent recruitment of SOMs, where there was an overall reduction in SOM activation, except at F-Type connections, where higher frequency transmission was actually enhanced (Fig. 4 and S8). In addition, visual deprivation led to a shift in connection probability to favor closer PC and SOM pairs (Fig. S9a), which may have implications for how SOMs integrate local information. Similar to that seen in L5 of juveniles[22], local connections in the adult V1 L2/3 display asynchronous

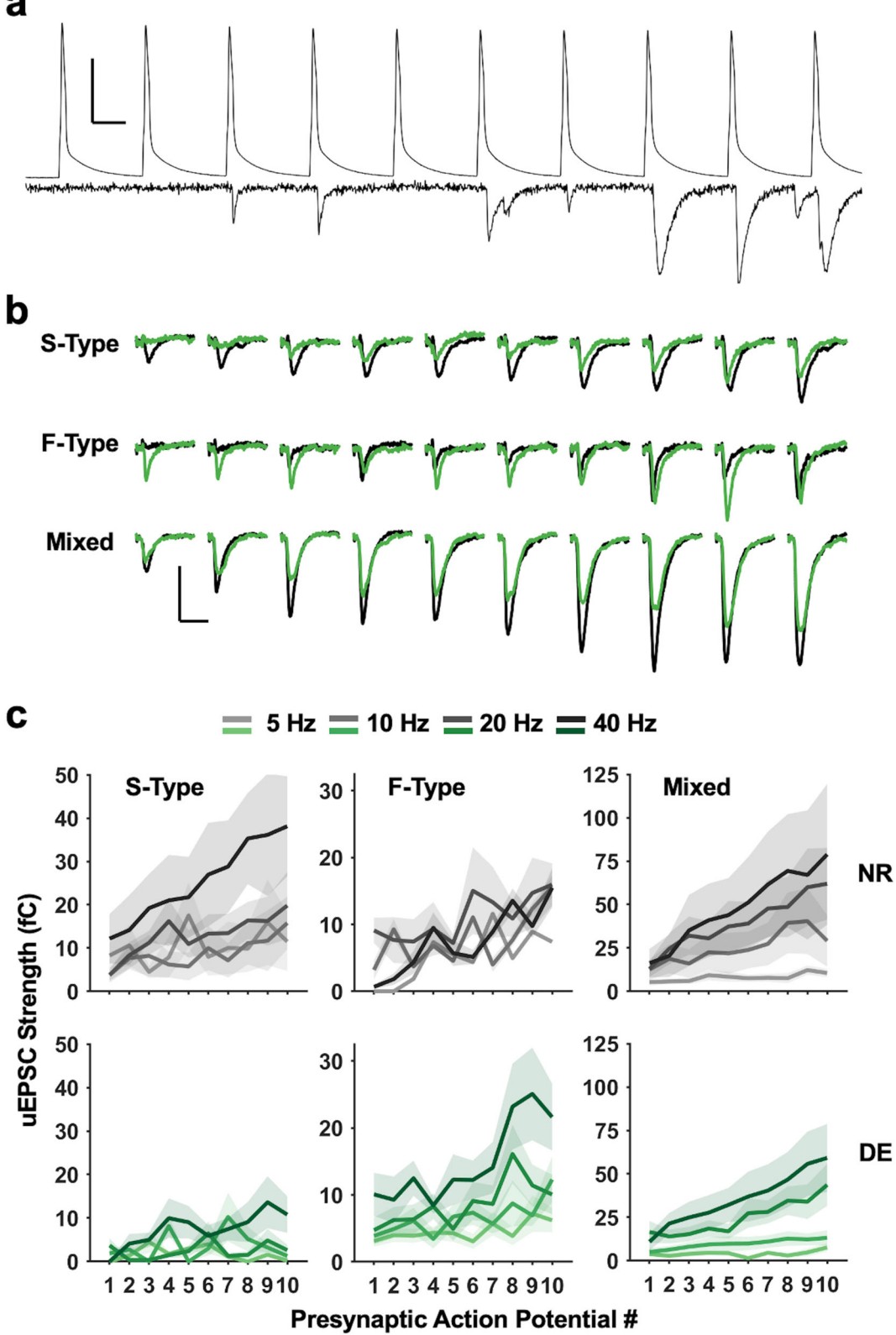

**Fig. 3 | Different types of local PC to SOM pair connections display short-term facilitation with varied properties and differential regulation by DE. a** An example current-clamp trace from a presynaptic PC (top) and a corresponding voltage-clamp trace recorded from a postsynaptic SOM (bottom). Scale bar: 50 mV (top: voltage trace), 50 pA (bottom: current trace), 10 ms. **b** Average uEPSC traces were recorded from SOMs in S-type pairs (top), F-Type pairs (middle), and Mixed pairs (bottom) with trains of 10 presynaptic action potentials evoked in the presynaptic PC at 40 Hz. Average uEPSC traces from NR (black) are overlaid with those recorded from DE (green). Scale bars: 10 pA, 10 ms. **c** Comparison of uEPSC strength for each presynaptic action potential in a train across different stimulation frequencies (5 Hz, 10 Hz, 20 Hz, and 40 Hz). Left panels: S-Type pairs. Middle panels: F-Type pairs. Right panels: Mixed pairs. Top row: Results from NR (gray). Bottom row: Results from DE (green). Lines: mean values. Shaded areas: standard error of the mean (S.E.M.). See Tables S1 and S2 for statistics.

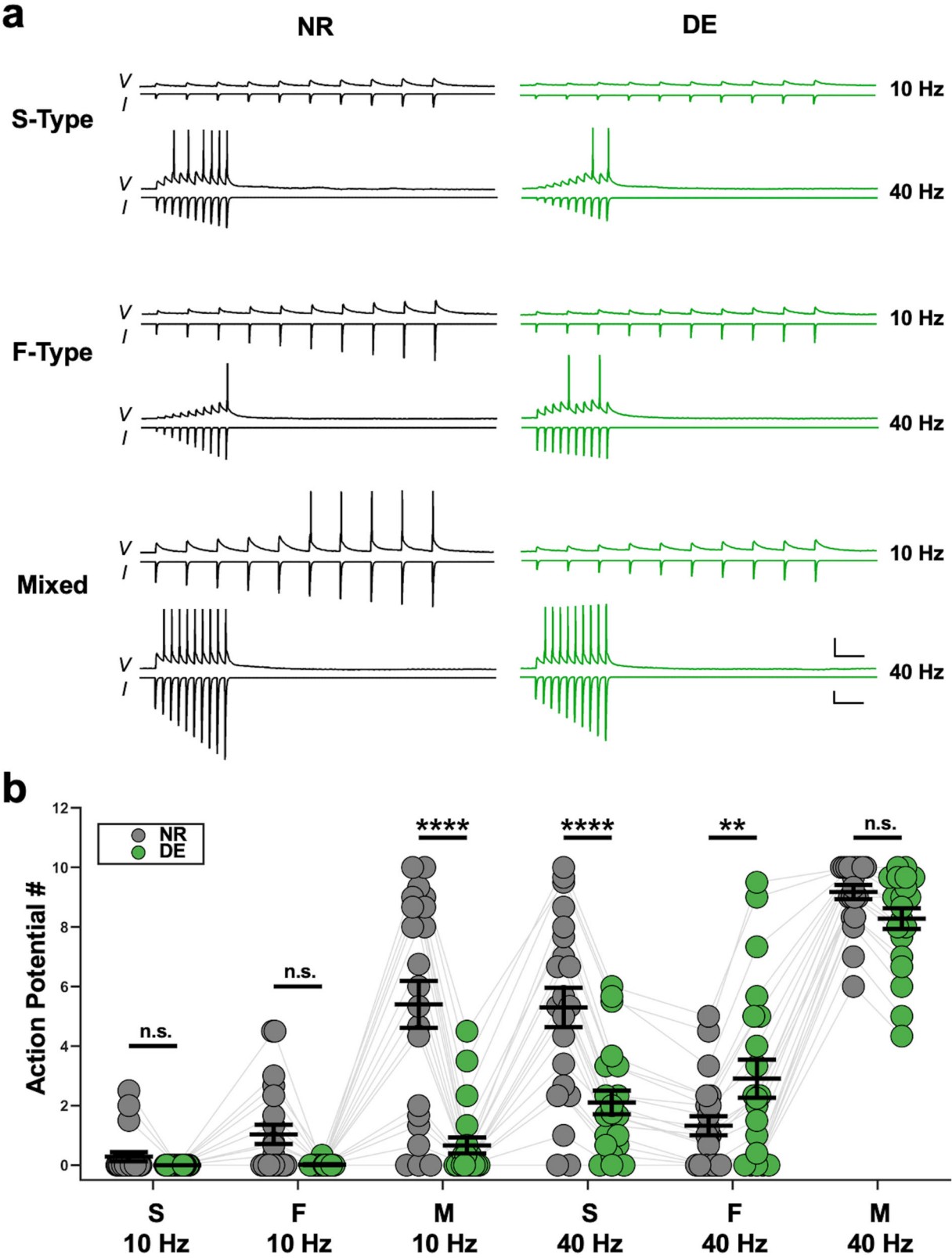

release which is not regulated by visual deprivation (Fig. 5). On the other hand, crossmodal auditory deprivation reduced the variance in mEPSC type proportion across cells and led to an increase in S-Type mEPSC frequency (Fig. 6), likely reflecting an increase in the number of S-Type synapses or baseline probability of release at S-Type synapses.

One of the major findings in our study is that V1 L2/3 SOMs contain excitatory synapses with two distinct AMPAR kinetics. These S- and

F-Type synapses co-exist on SOMs in what appears to be on a continuum of varying proportions (Fig. 1). The kinetics of F-Type mEPSCs are similar to those observed in PVs, while those of S-Type mEPSCs fall within the range seen in PCs (Fig. S10). We found that mEPSCs in L2/3 SOMs are reduced in amplitude with a CP-AMPAR antagonist (Fig. S4b). This distinguishes this class of MCs from those in L5, which do not express CP-AMPARs[30]. These results are consistent with the recent

**Fig. 4 | DE-induced plasticity of uEPSC short-term dynamics alters SOM activity. a** Average voltage traces (*V*) recorded in current clamp during playback of 10 and 40 Hz idealized traces (*I*). All traces are from the same control V1 L2/3 SOM cell. Top panels: Recordings from playback of idealized S-Type unitary connection current traces. Middle panels: Recording from playback of idealized F-Type unitary connection current traces. Bottom panels: Recording from playback of idealized Mixed connection current traces. Left column: Playback of idealized current traces from NR mice (black). Right column: Playback of idealized current traces from DE mice (green). Scale bars: *V* traces 25 mV, 100 ms; *I* traces 125 pA, 100 ms.
**b** Comparison of the number of action potentials generated in SOMs while playing back the different idealized current traces. All SOMs that were recorded from

(*n* = 21 cells from 5 mice) received playback of NR and DE traces for each frequency (10 and 40 Hz) and connection type (S-Type, F-Type, and Mixed), for a total of 12 unique conditions. Circles: average value from each cell. Thick black lines: group mean ± S.E.M. Thin gray lines connect data points obtained from the same SOM cell. Data were fit with a linear mixed effects model. Analysis of main effects and interaction terms was carried out using Type III ANOVA with Satterthwaite's method (Table S3). Further pairwise contrasts of estimated marginal means (Table S4), were computed using *z*-tests with Dunnett's correction for multiple comparisons. A subset of these pairwise contrasts (instances of NR versus DE) is indicated above the groups compared (****$p < 0.0001$, **$p < 0.01$, n.s.: not significant, see Table S4 for exact *p* values.).

classification of L2/3 and L5 MCs as distinct MET-types with different gene expression profiles[25]. In contrast to PVs and PCs, where EPSCs mediated by CP-AMPARs have faster kinetics[29,45], our data suggest that CP-AMPARs are predominantly found at S-Type synapses in SOMs (Fig. S4c). The association of CP-AMPARs with auxiliary subunit TARPs has been shown to slow down the kinetics of their currents by slowing deactivation and reducing desensitization[31]. This is especially prominent in assemblies containing the γ−4 or γ−8 TARP isoforms, which display currents with substantially slowed decay kinetics (3 − 4 fold increase in decay τ)[46,47] and increased rise times[47]. These CP-AMPAR complexes could contribute to the larger charge transfer seen in S-Type synapses (Fig. S3). SOMs express γ−4, among other TARPs[32], which supports our conclusion that S-Type synapses on V1 L2/3 SOMs contain CP-AMPARs with slow kinetics. Finally, the heterogeneous mEPSCs we observe in L2/3 SOMs are not restricted to adult V1 and are present in immature V1 (Fig. S5), suggesting that it may be an innate feature of these neurons. However, it may not be a property unique to L2/3 SOMs, as heterogeneous mEPSCs have also been observed in a subset of hypothalamic orexin neurons[48].

We found that about half of the local unitary connections from L2/3 PCs contain both S- and F-Type synapses (Fig. 2b). The notion that a single postsynaptic cell can possess synapses with different AMPAR kinetics is not new, but our findings contain important distinctions. It has been shown that certain synapses can undergo plasticity to recruit or remove CP-AMPARs in an activity-dependent manner (mostly reported for CP-AMPARs with faster kinetics)[45], which suggests that a major determinant for expressing AMPARs with different kinetics is the specific activity of distinct inputs. Our results differ from this idea, as we have observed that a unitary PC to SOM connection can contain both S- and F-Type synapses, which we have demonstrated to have distinct kinetics and short-term dynamics (Figs. 2 and 3). How this arises when the presynaptic neuronal activity is presumably the same at the different synapses within a single paired connection, is an interesting question to pursue in future studies. Our observation that there are pairs of connections that only contain S- or F-Type synapses, in addition to the ones harboring both types, suggests that the types of synapses utilized might be shaped by distinct needs of the local microcircuitry. Our finding that the composition of synaptic type shifts with changes in sensory experience (Fig. 6) supports this idea.

Our study suggests that S- and F-Type synapses have unique functions under normal conditions and undergo differential regulation by sensory experience. Under conditions of normal visual function, local V1 L2/3 PCs with Mixed connections play a dominant role in recruiting SOMs, and S-Type connections increase their contribution at higher activity levels (Fig. 4 and S8, Tables S3, S4, and S5). Following visual deprivation, recruitment of SOMs by local PCs with different connection types is reduced overall, except at F-Type connections under conditions of higher presynaptic PC activity. Also following visual deprivation, SOMs lose connections from more distal PCs (Fig. S9). Given that V1 L2/3 SOMs are implicated in providing surround suppression[8] and feedback modulation of PCs[9], the general reduction in SOM recruitment following visual deprivation suggests L2/3 PCs may respond better to information from outside of their classical

receptive fields. In contrast, plasticity of SOMs following auditory deprivation differs in that it enhances the contribution of S-Type synapses (Fig. 6), which may serve a purpose for adapting the V1 circuit to crossmodal sensory loss. Because S-Type synapses are more effective at recruiting SOMs, increasing S-Type contribution could provide better suppression of PC responses by contextual information and thus may enhance visual processing. In addition, we reported previously that AD of adult mice accelerates ocular dominance plasticity by restoring LTP of thalamocortical inputs to V1[49]. It is known that SOMs provide inhibition not only to PCs but also to PVs[50,51]. Transient disinhibition of the PV circuit is considered an initial step that enables ocular dominance plasticity during the early postnatal critical period[19,20], and a developmental decrease in SOM inhibition of PVs is implicated in the closure of the critical period[10]. Increasing SOM recruitment as expected following AD is predicted to enhance the inhibition of PV neurons, which would result in the disinhibition of local PCs to enable plasticity. We surmise that the increase in the proportion of S-Type synapses on SOMs by crossmodal AD may provide the transient disinhibition of adult V1, which is necessary to initiate plasticity.

Collectively, our results demonstrate that SOMs in V1 L2/3 are poised for exquisite regulation by utilizing two different types of excitatory synapses, originating even, from the same presynaptic neuron. In adults, these two types of excitatory synapses undergo independent, differential regulation by changes in sensory experience, which we expect fine-tunes V1 functionality to adapt to sensory perturbations.

## Methods

### Mice
Male and female Tg(GadGFP)45704Swn (GIN; The Jackson Laboratory, Stock #: 003718) mice were reared in a 12 h light/12 hr dark cycle. The mouse facility is maintained at ambient temperature 20 − 25 °C and 30 − 70% relative humidity. Adult animals were dark exposed (DE) for 0 or 7 days, with DE beginning between P90 and P113 and recording taking place between P90 and P120. DE animals were cared for in a dark room with infrared vision goggles using dim infrared light. Young animals were used between P14 − P16. All experiments were done in accordance with protocols approved by Institutional Animal Care and Use Committee of Johns Hopkins University.

### Acute slice preparation for mEPSC recording
Mice were anesthetized with an overdose of isoflurane vapors and decapitated after verifying the absence of a toe-pinch response. Brain blocks containing visual cortex were coronally sliced into 300-μm sections using a vibratome (Pelco EasiSlicer™) in ice-cold dissection buffer containing 212.7 mM sucrose, 10 mM dextrose, 3 mM MgCl₂, 1 mM CaCl₂, 2.6 mM KCl, 1.23 mM NaH₂PO₄•H₂O, and 26 mM NaHCO₃, which was bubbled with a 95% O₂/5% CO₂ gas mixture. Slices from young mice were incubated at room temperature for 60 min in artificial cerebrospinal fluid (ACSF: solution containing 124 mM NaCl, 5 mM KCl, 1.25 mM NaH₂PO₄•H₂O, 26 mM NaHCO₃, 10 mM dextrose, 2.5 mM CaCl₂, and 1.5 mM MgCl₂, bubbled with 95% O₂/5% CO₂) and slices from

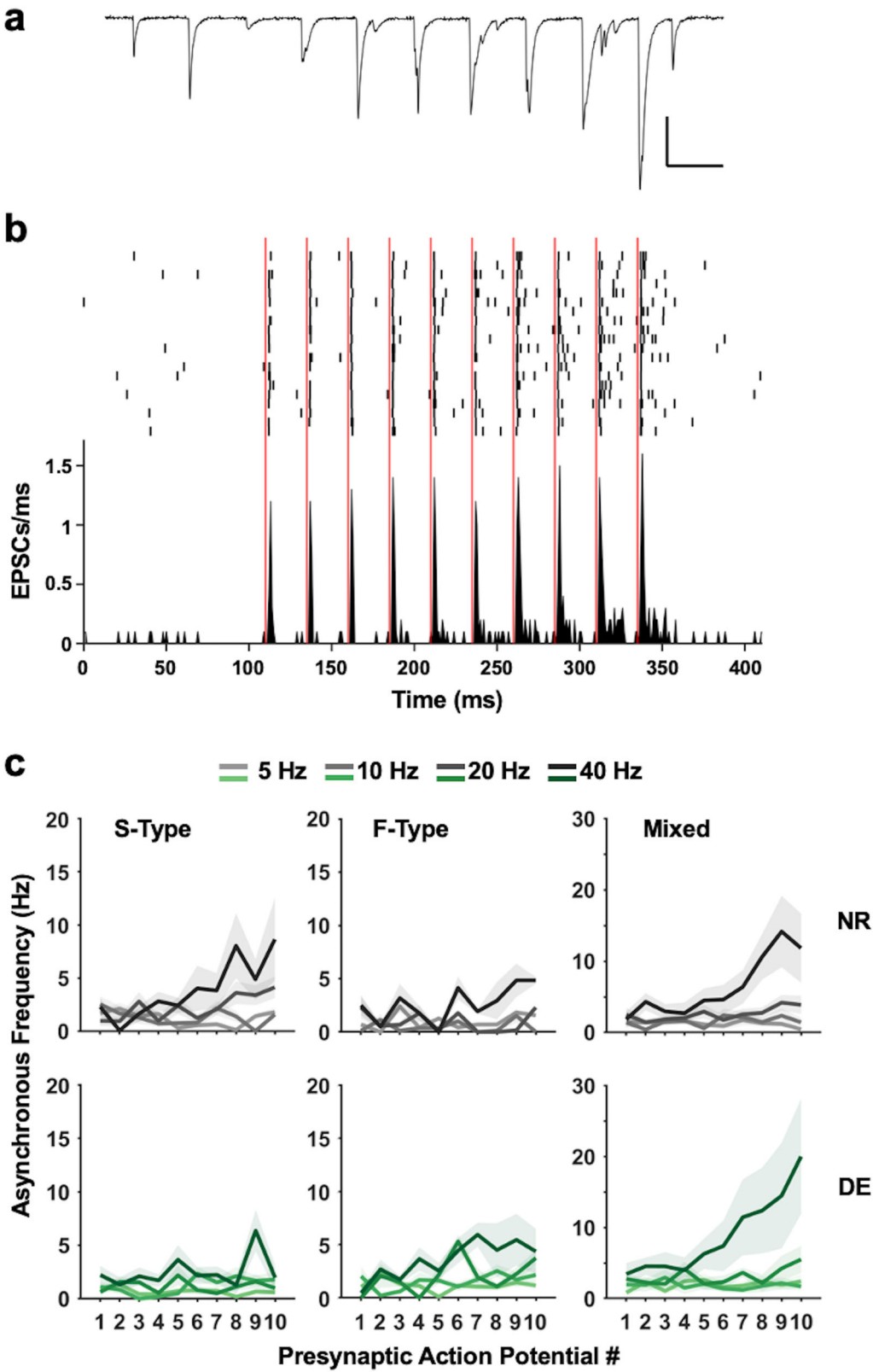

**Fig. 5 | Asynchronous release occurs at 3 types of local PC to SOM connections and is not regulated by visual experience. a** An example current trace recorded from a SOM showing asynchronous release during a train of stimulation (40 Hz). Note increasingly prominent asynchronous events with increasing activity. Scale bars: 100 pA, 25 ms. **b** Quantification of asynchronous release. Top: Raster plot of EPSCs from an example pair. Each row represents a sweep of 10 stimulations. Vertical red lines depict the time of a presynaptic action potential peak. Bottom: Peristimulus time histogram from example pair. Note the increased prevalence of asynchronous EPSCs following action potentials later in the train. **c** Comparison of asynchronous release rate for each presynaptic action potential in a train across different stimulation frequencies (5 Hz, 10 Hz, 20 Hz, and 40 Hz). Left panels: S-Type pairs. Middle panels: F-Type pairs. Right panels: Mixed pairs. Top row: Results from NR (gray). Bottom row: Results from DE (green). Lines: mean values. Shaded areas: S.E.M.

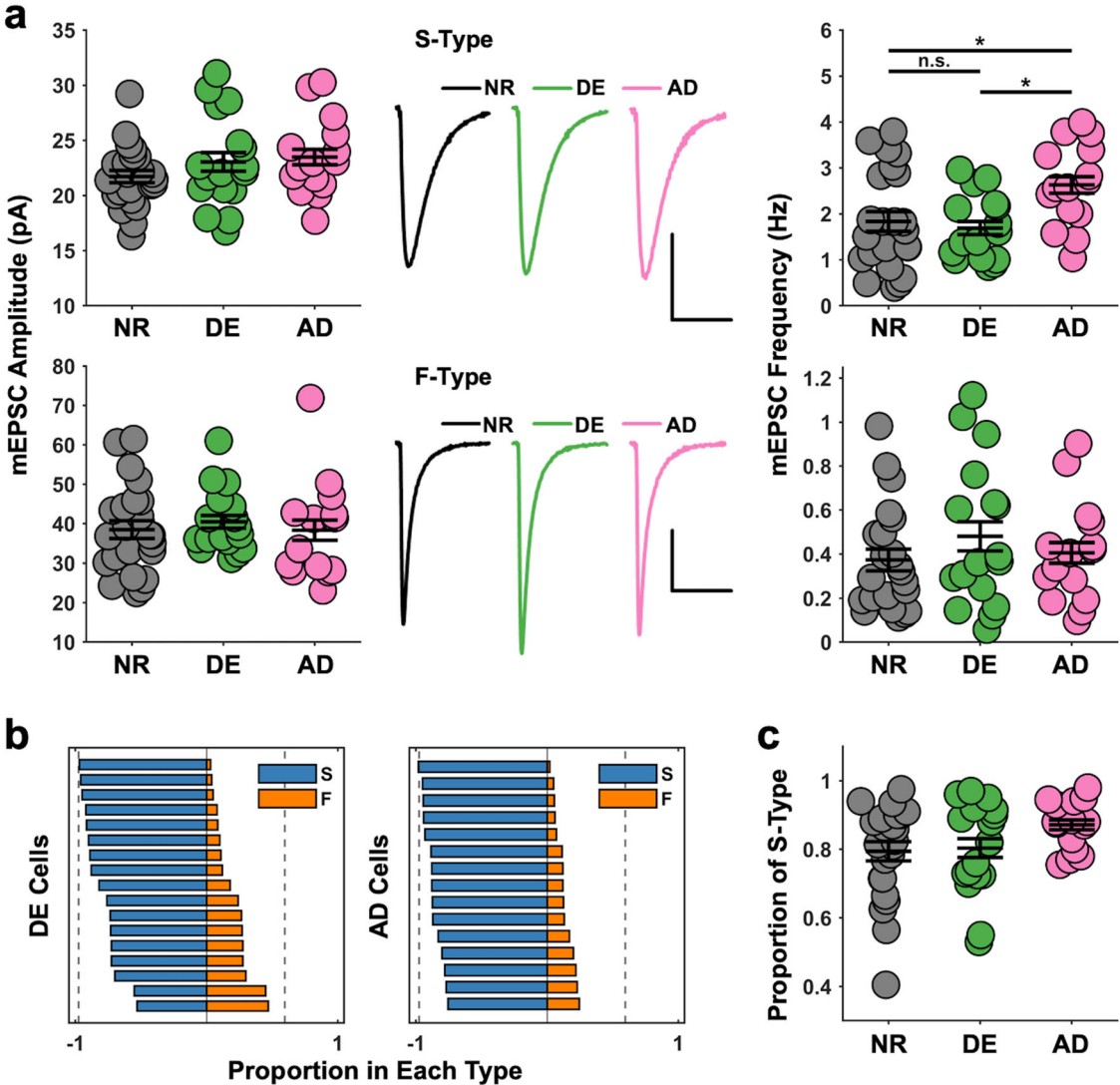

**Fig. 6 | DE does not alter mEPSCs, while AD increases the frequency of S-Type mEPSCs. a** S-Type mEPSCs are selectively regulated by AD. Top left: Changes in sensory experience do not affect S-Type mEPSC amplitude (One-way ANOVA: F = 1.36, p = 0.2648). Top center: average S-Type mEPSCs (scale bar: 10 pA, 10 ms). Top right: Following AD, the frequency of S-Type mEPSCs is significantly greater than in the NR or DE conditions (One-way ANOVA: F = 4.66, *p = 0.0139; Tukey's HSD NR-DE: p = 0.8831; Tukey's HSD NR-AD *p = 0.0346, Tukey's HSD DE-AD *p = 0.0193). Bottom left: Changes in sensory experience do not affect the amplitude of F-Type events (One-way ANOVA: F = 0.21, p = 0.8096). Bottom center: average F-Type mEPSCs (scale bar: 10 pA, 10 ms). Bottom right: Changes in sensory experience do not affect F-Type mEPSC frequency (One-way ANOVA: t = 0.77, p = 0.4664). Circles: average data point for each cell. Thick black lines: group mean ± S.E.M. Sample size: NR, n = 23 cells; DE, n = 17 cells; AD, n = 15 cells. **b** of the proportion of S- and F-Type mEPSCs in each SOM cell from the DE (left panel) and AD (right panel) groups. Each bar represents a cell. Dotted lines represent the bounds of proportions in adult NR SOM cells. **c** The variance in the proportion of S-Type mEPSCs across cells is decreased following AD (Bartlett's test: $\chi^2 = 6.91$, *p = 0.0316). Circles: average data point for each cell. Thick black lines: group mean ± S.E.M. Sample size: NR, n = 23 cells; DE, n = 17 cells; AD, n = 15 cells.

adult mice were incubated in ACSF for 30 min at 30 °C, followed by 30 min at room temperature.

## Whole cell voltage-clamp recordings of mEPSCs

Visual cortical slices were transferred to a submersion-type recording chamber mounted on the fixed stage of an upright microscope with oblique infrared illumination and were continually supplied with 2 ml/min of 30 °C ACSF bubbled with 5% $CO_2$/95% $O_2$. AMPA receptor-mediated miniature excitatory postsynaptic currents (mEPSCs) were isolated by adding 1 μM tetrodotoxin, 20 μM bicuculline, and 100 μM DL-2-amino-5-phosphonopentanoic acid. Recording pipettes were filled with an internal solution containing: 130 mM Cs-gluconate, 10 mM HEPES, 8 mM KCl, 1 mM EGTA, 4 mM $Na_2$•ATP (Sigma-Aldrich Cat# A6419), 10 mM $Na_2$•phosphocreatine (Sigma-Aldrich Cat# P7936), 0.5 mM Na•GTP (Sigma-Aldrich Cat# G8877), 5 mM lidocaine N-ethyl

bromide (Sigma-Aldrich Cat# L5783). In a subset of experiments, 20 μM 1-naphthyl acetyl spermine trihydrochloride (Naspm; Tocris Cat# 2766) was included to block calcium-permeable AMPARs. In an additional experiment, 10 μM 2,3-Dioxo-6-nitro-1,2,3,4-tetrahydrobenzo[f]quinoxaline-7-sulfonamide disodium salt (NBQX) was bath applied to block glutamatergic currents. Somatostatin (SOM) interneurons in L2/3 of the primary visual cortex (V1) were positively identified through the expression of green fluorescence protein (GFP) and were recorded in voltage clamp at −80 mV. The recorded mEPSCs were digitized at 10-kHz by a National Instruments data acquisition board and acquired through a custom program written in Igor (Igor Pro v.8.0, Wavemetrics; program available at https://github.com/heykyounglee/4xLTP-igor and https://doi.org/10.17632/42p6m5638j.1). SOM cells were excluded from analysis if series resistance ($R_s$) was greater than 30 MΩ. Parvalbumin interneurons (PVs) and pyramidal

cells (PCs) were excluded if $R_s$ were greater than 25 MΩ. Additionally, cells were excluded if $R_s$ changed by greater than 15% over the course of the recording. At least 200 well-isolated mEPSCs were analyzed from each cell using a custom program written in Matlab (R2020b, Mathworks). This program is freely available online (https://github.com/bdgrier/whole-cell-data-analysis).

### Whole cell voltage-clamp recordings of evoked unitary excitatory postsynaptic currents

Visual cortical slices were transferred to a submersion-type recording chamber mounted on the fixed stage of an upright microscope with oblique infrared illumination and were continually supplied with 2 ml/min of 30 °C ACSF bubbled with 5% $CO_2$/95% $O_2$. AMPA receptor-mediated unitary excitatory postsynaptic currents (uEPSCs) were isolated by 20 μM bicuculline and 100 μM DL-2-amino-5-phosphonopentanoic acid (APV). Recording pipettes were filled with an internal solution containing: 130 mM K-gluconate, 10 mM HEPES, 20 mM KCl, 0.3 mM EGTA, 4 mM $Na_2$•ATP, 10 mM $Na_2$•phosphocreatine, 0.5 mM Na•GTP. Postsynaptic somatostatin interneurons in L2/3 of V1 were positively identified through an expression of GFP and were recorded in voltage clamp at −80 mV. Presynaptic PCs were identified by a triangular shape and prominent apical dendrite. Trains of 10 presynaptic action potentials (APs) were generated every 10 s at 5, 10, 20, or 40 Hz with a 2 nA stimulus applied for 2 ms. The recorded action potentials and uEPSCs were digitized at 10-kHz by a National Instruments data acquisition board and acquired through a custom program written in Igor (Igor Pro v.8.0, Wavemetrics; program available at https://github.com/heykyounglee/4xLTP-igor and at https://doi.org/10.17632/42p6m5638j.1). Pairs were excluded from uEPSC analysis if the SOM $R_s$ were greater than 40 MΩ. For a given pair, 10-20 sweeps were analyzed for each frequency. uEPSCs were analyzed using a custom program written in Matlab (R2020b, Mathworks) that is freely available online (https://github.com/bdgrier/whole-cell-data-analysis).

### EPSC clustering and classification

To cluster mEPSCs and sEPSCs, we first represented events as points in two dimensions (*ln(amplitude)* in the first dimension and *ln(charge transfer)* in the second). We then developed a cost function to determine the optimal angle of rotation for the data. This function was defined as: *argmax((a(θ)·c(θ))+(b(θ)·c(θ))):for* $0 \leq θ \leq min(c)$, where *a* represents the modal interval in dimension 2 (as determined from Hartigan's dip test[52]), *b* represents the range in dimension 1, and *c* represents the Akaike information criterion of a two-component Gaussian mixture model (GMM) fit along dimension 2. The value of *θ* that minimized *c* with respect to *a* and *b* was selected as the optimal rotation angle. A two-component GMM was fit along dimension 2 of the rotated data, and events were hard clustered with a posterior probability (*p*) threshold of 0.5. The GMMs fit to sEPSCs were then used to classify uEPSCs from the same SOM cell, thus providing an internal basis that did not necessitate the presence of two types of events in the uEPSCs.

### Connectivity analysis

Following each paired recording, separate images were taken of the presynaptic and postsynaptic recording pipette tips in focus. The z-distance between the two cells was measured from the number of gradations on the fine focus of the microscope between the in-focus planes of each pipette tip. The x- and y- distances were measured *post hoc* using the acquired images. To prevent false negatives due a low probability of release at the investigated synapses, connectivity was determined using trains of 10 presynaptic APs at 40 Hz.

### Analysis of short-term dynamics and asynchronous release frequency

In the paired whole-cell recording configuration, trains of 10 presynaptic APs were elicited at 5, 10, 20, and 40 Hz. In volage-clamp,

measurements of success rate, strength, and potency were made on postsynaptic EPSCs that occurred within 3 ms following the peak of a presynaptic action potential. Asynchronous release frequency for a given action potential was quantified as the frequency of events beginning at 3 ms from the peak of the action potential and ending at the beginning of the following action potential.

These experiments generated multi-factor, repeated measures data, which were analyzed by first fitting linear mixed effects (LME) models (R package 'lme4'). The models in Table S1 were described by: *Parameter ~ 1 + AP\*Freq + (1|Cell)*. The models in Table S2 were described by: *Parameter ~ 1 + AP\*Freq + AP\*Type + AP\*Exp + Freq\*Type + Freq\*Exp + Type\*Exp + (1|Cell)*. In Table S3, the model in the "Ideal" column was described by: *Parameter ~ Type\*Exp\*Freq + (1|Cell)*, and the "Average" column was described by: *Parameter ~ Type\*Exp + (1|Cell)*. The main effect and interaction terms for LME models were tested for significance with F-tests (R package 'lmerTest'). Relevant contrast matrices were made for contrasts in Tables S2, S4, and S5, and contrasts of estimated marginal means were made using the R package 'multcomp'. Power analysis of LME model terms was computed using the R package 'simr'.

### Generation of idealized current traces

Idealized uEPSC traces were constructed based on the average strength, kinetics, and short-term dynamics of the uEPSCs from the paired recording data (PC to SOM) from NR and DE mice. Specifically, the series of average strength values for a given set of traces (e.g. NR + F-Type + 10 Hz) was fit with a linear model, and the coefficients of these models were used to generate a series of linearly increasing strength values for each set of traces. To determine kinetics, a beta function[53] was first fit to the average uEPSC for each set of traces. The resulting fit was then repeated and scaled according to its respective series of linearly increasing strength values, thus creating a trace with a series of linearly increasing synaptic currents.

### Current-clamp recordings with playback of idealized unitary ESPC traces

Idealized uEPSC traces generated from S-Type (S), F-Type (F), and Mixed (M) pairs, under NR and DE conditions, at 10 and 40 Hz presynaptic stimulation frequencies were used. Whole-cell current-clamp recordings were made from SOM-GFP neurons located in V1 L2/3 of normal-reared adult mice (~P90) using K-gluconate internal solution (130 mM K-gluconate, 10 mM KCl, 10 mM HEPES, 0.2 mM EGTA, 0.5 mM $Na_3$•GTP, 4 mM Mg•ATP, and 10 mM Na•phosphocreatine; pH 7.2-7.4, 280-290 mOsm). 100 μM APV, 10 μM NBQX, and 10 μM gabazine were added to ACSF to block synaptic transmission. The average resting membrane potential of SOM-GFP neurons was −63.5 ± 0.59 mV and the average input resistance was 400 ± 26.6 MΩ (*n* = 21 neurons from 5 mice). Idealized current traces were played back to SOM-GFP neurons under current clamp using the NeuroMatic (v.3.0) module in Igor Pro software (v.8.0, Wavemetrics)[54]. The gain of the playback was set at 25x to yield a range of action potential firing across NR and DE traces. At a playback gain of 1x, we did not observe action potentials with any of the playback current traces. A total of 12 unique idealized traces (S/F/M x 10/40 Hz x NR/DE) were played back 3 or 4 times in a pseudorandom order to each SOM neuron to obtain an average number of action potentials evoked by each trace. In another set of experiments, the average current traces from S-Type, F-Type, and Mixed pairs under NR and DE conditions at 40 Hz were played back to control SOM-GFP neurons (*n* = 12 cells from 4 mice). Spike detection was done using the NeuroMatic module (v.3.0)[54].

### Statistics

All mEPSC comparisons are displayed as mean ± SEM. Two-sample *t*-tests and paired-sample *t*-test were used for comparisons of the means of two groups. The means of 3 groups were compared using one-way

ANOVA. Normality was assumed for these comparisons. Short term dynamics are displayed as mean ± SEM for display purposes only and were compared as described above. Connectivity is displayed as a moving mean for display purposes only and was compared using logistic regression. Statistical analyses were performed in Matlab (R2022b, Mathworks), except where stated otherwise.

## Reporting summary
Further information on research design is available in the Nature Portfolio Reporting Summary linked to this article.

## Data availability
Data reported in this study are publicly available at Mendeley Data (https://doi.org/10.17632/gpg8jnctrn.1) and provided in the Source Data file. Source data are provided with this paper.

## Code availability
The custom Igor Pro (v8.0, Wavemetrics) data acquisition program is publicly available on the Lee Lab Github (https://github.com/heykyounglee/4xLTP-igor) and at Mendeley Data (https://doi.org/10.17632/42p6m5638j.1). The whole-cell analysis program is publicly available on Bryce Grier's Github (https://github.com/bdgrier/whole-cell-data-analysis).

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

## Acknowledgements

This work was supported by the National Institutes of Health (NIH) grants R01-EY014882 to H-KL and F31-EY031946 to SP.

## Author contributions

B.D.G. performed all of the experiments and analyses, except for recordings of current playback (S.P. and H.-K.L.) and confocal imaging (J.O. and S.P.). B.D.G. and H.-K.L. conceptualized the study and wrote the manuscript.

## Competing interests

The authors declare no competing interests.
