## [Peer Review File · Nature Communications]

Selective plasticity of fast and slow excitatory synapses on somatostatin interneurons in adult visual cortexREVIEWER COMMENTS

Reviewer #1 (Remarks to the Author):

The manuscript by Grier and Lee describes experiments making electrophysiological recordings from somatostatin interneurons in adult visual cortex layer 2/3, and report that even between a single presynaptic and single postsynaptic neuron, EPSCs can fall into two distinct kinetic categories. This finding is novel, and suggests that two very different types of synapses occur on these neurons. The authors then go on to a careful analysis of how each type of synapse (fast, slow, and mixed) exhibits short term plasticity, asynchronous transmitter release, and is modified after the mice are kept in the dark for 7 days or are deprived of auditory input. Overall, these observations support the conclusion that distinct synapses onto somatostatin interneurons exist and are differentially affected under a variety of biologically relevant conditions. The data are of high quality, and the analysis and interpretation are carefully done. The findings as presented may be of more interest to those within the field of cortical circuits and visual cortex adaptations.

Minor suggestions for revision:

1. It would be of interest to speculate on what the mechanisms are that account for the reported differences in synaptic function, as well as how these individual synaptic properties may be well-suited to controlling aspects of visual cortical function.
2. What do the authors think is the function of mixed rather than F or S type synapses? As the proportions can change with experience, what does this mean for the circuit?
3. Figure 4C should be labeled with the frequencies used (as in 3).

Reviewer #2 (Remarks to the Author):

This manuscript reports the kinetics and plasticity of excitatory postsynaptic currents (EPSCs) recorded from L2/3 SOM+ interneurons in the visual cortex. Through electrophysiological recording, the authors identified two types of EPSCs based on their kinetics, which respond differently to environmental changes. Specifically, visual deprivation mainly affects fast kinetics EPSCs, whereas auditory deprivation selectively regulates slow kinetics EPSCs. These findings shed new light on our understanding of cortical message processing. However, the study comes across as preliminary and phenomenological in current version, and there are some methodological limitations that need to be addressed.

1. The quality of EPSC recording largely depends on series resistance. A large series resistance can slow down the kinetics of EPSCs and disrupt their classification into fast- and slow-kinetics types. Is there any special reason for using different series resistance thresholds for different cell types? And even for paired recording (harder to achieve), 40 M Ω is too high.
2. Evidence supporting the presence of CP-AMPA receptors at excitatory synapses on L2/3 SOM+ interneurons was not sufficient. The small difference in mEPSC amplitude may have resulted primarily from long-term recording after the Naspam application. Furthermore, it will be clearer to show the respective change of fast- or slow- type mEPSC here.
3. The method of visual deprivation used in the study involved placing adult mice in the dark for seven days. Is it an appropriate method for inducing visual deprivation and whether it causes behavioral alteration? Normally, visual deprivation is performed during the critical period (P20-P35) by using monocular deprivation. What plasticity would happen if using the canonical method?
4. Asynchronous release (AR) events generally increase with stimulus number due to residual calcium accumulation, making AR events rare after the initial stimulus. However, the initial AR frequency is around 5 Hz for all types of PC-SOM pairs in Fig. 4c. Is this due to the count of baseline EPSC events? If so, the baseline frequency should be removed.
5. What is the role of different types of EPSC in shaping the action potential generated in SOM+ interneurons? Such as the firing probability or precision. The circuit function of different types EPSCs remains unknown.
6. The original source of different types of EPSCs remains unclear. In addition to input differences, could subset differences of postsynaptic SOM+ cells contribute to distinct EPSC kinetics?
7. What is the relationship between baseline frequency and types of mEPSCs? For fast-type

synapses, whether they hold higher baseline frequency thus causing uEPSC could not be distinguished from mEPSC. This could lead to the finding that the synaptic strength of F-type uEPSCs was only sensitive to stimulus number but not stimulus frequency. Because longer stimulation times involved more mEPSCs, ultimately leading to an increase in synaptic strength.

8. Please give the full name when an abbreviation was used for the first time in the manuscript. (e.g., CP-AMPA)

9. Please ensure that all figure panels and corresponding legends are thoroughly reviewed to confirm their accuracy. (e.g., Fig. S6b)

Reviewer #3 (Remarks to the Author):

Selective plasticity of fast and slow excitatory synapses on somatostatin interneurons in adult visual cortex

Grier and Lee show that excitatory synapses of somatostatin expressing inhibitory neurons (SOM) in the primary visual cortex (V1, layer 2/3) can be categorized into fast (F) and slow (S) types based on their AMPAR kinetics. Both types of synapses are identifiable when measuring both spontaneous mEPSCs and unitary events, indicating a functional integration of both types in the local circuitry. Notably, F and S types of synapses are present in both the juvenile and the adult V1 and show distinct plastic adaptation following unimodal visual or cross-modal auditory deprivation.

The data presented in this study is interesting, as they further our understanding of the intricate manner in which SOM neurons are integrated into local circuitry in V1. Additionally, the research sheds light on how these neurons' excitatory synapses adapt differently to distinct alterations in sensory experiences. While I am not an expert in this field, I believe this research paper is beautifully crafted and exhibits clarity in its writing. Hence, I only have a few comments (see below). Specifically, I noticed that the authors do not address whether their significant finding is unique to SOM cells or represents a common characteristic of excitatory synapses across various cell types in layer 2/3 of V1. To ensure the comprehensiveness required for a Nature Communications publication, it would be valuable to experimentally explore this aspect as well (see Major comment 1).

Major comment

1. The authors provide compelling evidence for the existence of F and S excitatory synapses on SOM inhibitory neurons. However, it remains unclear whether this phenomenon is unique to SOM cells or if it can be observed in other neuronal types, such as pyramidal neurons or parvalbumin (PV) positive inhibitory neurons, as well. Existing literature suggests that pyramidal neurons may exhibit similar categories of excitatory synapses as SOMs. To investigate this further, the authors could examine spontaneous mEPSCs in pyramidal and PV neurons in layer 2/3 of V1, applying the same analysis as performed for SOM neurons in Figure 1. These results could offer preliminary insights into the presence or absence of analogous excitatory synapse populations in different cell types and help determine whether this finding is a general feature of excitatory synapses or unique to SOMs in layer 2/3 of V1.

Minor comments

2. In the introduction, the authors highlight the "critical role of SOM" (and VIP) inhibitory neurons in cross-modal sound-mediated receptive field sharpening in V1, citing Ibrahim et al., 2017 as evidence. However, the cited study does not establish a crucial role for SOM neurons in cross-modal multisensory processing, as it states that "neither PV nor SOM neurons appeared to contribute significantly to the sound-induced effect." Furthermore, the paper does not present data on cross-modal receptive field sharpening, but instead investigates the impact of sounds on orientation selectivity of individual neurons. Although the orientation selectivity of a neuron is closely related to its receptive field, the authors should (in my opinion) interpret this finding more carefully. As such, the authors should consider rephrasing this statement or emphasizing the potential importance of SOM neurons in multisensory processing from a different perspective.

3. It would be beneficial to depict an image of a brain slice containing V1, in which MCs expressing GFP are visible, in either Figure 1 or Supplementary Figure 1. This addition would make the related Figure more intuitive, allowing readers to immediately identify the cell type of interest in this study.

4. In Figure S4b, the authors show that Naspm application substantially diminishes mEPSC amplitudes in SOM neurons on a global scale, which suggests the existence of CP-AMPA receptors in their membranes. To determine if Naspm preferentially affects mEPSC amplitudes of either S-type or F-type excitatory synapses, the authors could analyze the data separately for each synapse type. This distinction would offer preliminary evidence regarding whether CP-AMPA receptors are predominantly involved in either S-type or F-type responses.

Response to Reviewers' Comments

We thank the reviewers for assessing our work to be “novel,” “shed[ding] new light,” and “further[ing] our understanding of the intricate manner in which SOM neurons are integrated into local circuitry in V1.” While the reviewers agreed on the significance and novelty of our findings, they made several constructive comments that can improve our interpretation of data, address the methodological limitations, and add comprehensiveness to our study that can increase and broaden the impact. As detailed below, we have carefully addressed all the comments by adding new analyses and experiments, as well as clarifying our narrative by revising the text and figures. The new texts are highlighted in blue font in the marked-up version of the text. We also include a clean copy without the mark ups.

Reviewer #1

We thank the reviewer for pointing out that “[t]he data are of high quality, and the analysis and interpretation are carefully done.”

The reviewer made the following 3 minor suggestions:

1. It would be of interest to speculate on what the mechanisms are that account for the reported differences in synaptic function, as well as how these individual synaptic properties may be well-suited to controlling aspects of visual cortical function.

We thank the reviewer for the suggestion. Based on comments from Reviewer #2, we now added a new set of experiments to see how the changes in synaptic function may affect activity propagation to SOMs (see our response to Reviewer #2 comment #5 for details). Analysis of this new data set suggests that unitary connections from local pyramidal cell (PC) containing both S- and F-Type synapses (Mixed connections) are much better at driving action potentials in SOMs compared to S-Type only or F-Type only connections (new Figures 5 and S8; statistics in Table S3, S4, and S5). Furthermore, we found that both Mixed and S-Type connections exhibit higher activity transmission at higher frequency synaptic inputs, while F-Type only connections did not. We interpret these results to suggest that local PCs utilizing Mixed connections will most likely recruit SOMs, and those using S-Type only connections would contribute to additional recruitment when there is higher activity in PCs. While the contribution of F-Type only local connections in driving SOM activity is likely minimal under normal conditions, their relative contribution increases following visual deprivation suggesting that they may play a rather unique role in adapting V1 to visual deprivation. These details are now added to our revised manuscript. The exact nature of how these circuit-level plasticity contribute to visual processing or adaptation to vision loss would require follow up studies using *in vivo* tools.

2. What do the authors think is the function of mixed rather than F or S type synapses? As the proportions can change with experience, what does this mean for the circuit?

As mentioned above, we have new data supporting the idea that Mixed connections are the most effective at recruiting SOMs based on activity of local PCs. Also, visual deprivation-

induced changes in short-term dynamics of the 3 types of PC to SOM connections, suggest that the recruitment of SOM activity overall would decrease, except for the contribution of F-Type connections at higher frequency synaptic activities. The increase in the proportion of S-Type synapses following auditory deprivation would suggest that it would act to boost recruitment of SOM activity by local PCs. Since one of the proposed roles of SOMs in visual processing is surround suppression, it is tempting to speculate that the plasticity observed following auditory deprivation would increase SOM recruitment by local PCs to facilitate surround suppression thus enhancing visual processing. It would be interesting to confirm such predictions in future studies.

3. Figure 4C should be labeled with the frequencies used (as in 3).

We added the frequency information in the referenced figure, which is now Fig. 5C in the revised manuscript.

Reviewer #2

1. The quality of EPSC recording largely depends on series resistance. A large series resistance can slow down the kinetics of EPSCs and disrupt their classification into fast- and slow-kinesis types. Is there any special reason for using different series resistance thresholds for different cell types? And even for paired recording (harder to achieve), 40 M Ω is too high.

We agree that high series resistance (R_s) can affect the voltage-clamp conditions, but considering that the input resistance of SOMs in adult V1 L2/3 is quite high (~ 400 M Ω), the expected error in voltage clamp is $\leq 10\%$ with the 40 M Ω cut-off. We believe the higher R_s for SOMs were not due to our patch-clamp methods, because even for paired recordings we consistently obtained lower R_s in PCs (< 25 M Ω). Hence, we believe the higher R_s for SOMs may be caused by biological differences in plasma membrane or extracellular matrix around these cells.

To confirm that the high cut-off was not affecting our current measurements, we ran correlation analyses between R_s and various parameters of EPSC measurements (see Figure provided here). We did not find any statistically significant correlations.

R_s cut-off of 40 M Ω for SOM cell recordings is not unique to our lab. We list below a few references from various groups using the same

cut-off. We would like to point out that the age ranges used in these labs are much younger (noted in parentheses) than ours (P90~120). These published studies support our idea that the higher R_s in SOMs could be of biological nature.

- Gord Fishell's group:
Wu et al. (2023) *Neuron*. <https://doi.org/10.1016/j.neuron.2023.05.032> (Age: P25-35)
Ibrahim et al. (2023) *eLife*. <https://doi.org/10.7554/eLife.86842> (Age: P18-22)
- Erika Fanselow's group:
Kinnischtzke et al. (2014) *Cereb Cortex*. <https://doi.org/10.1093/cercor/bht085> (Age: P31-51)
- David Prince's group:
Halabisky et al. (2005) *J Neurophysiol*. <https://doi.org/10.1152/jn.01079.2005> (Age: P28-35)
– This paper reports that the average R_s of SOMs in GIN mice is between 29~57 M Ω .

2. Evidence supporting the presence of CP-AMPA receptors at excitatory synapses on L2/3 SOM+ interneurons was not sufficient. The small difference in mEPSC amplitude may have resulted primarily from long-term recording after the Naspam application. Furthermore, it will be clearer to show the respective change of fast- or slow- type mEPSC here.

The duration of the recordings for the Naspam experiment was equivalent to the control experiment, because these were done in different neurons to avoid the potential “wash-out” problem as pointed out by the reviewer. We now clarify this in the figure legend.

We thank the suggestion of the reviewer for asking us to analyze Naspam effect on S- and F-type mEPSCs, which led to a rather interesting finding. After clustering the mEPSCs, we found that the reduction of mEPSC amplitude with Naspam was quite selective to S-Type mEPSCs without significant change in the amplitude of F-Type mEPSCs (see new Fig. S4c). This suggests that S-Type synapses contain slow CP-AMPA receptors, which is rather unusual. Prior studies demonstrated that the kinetics of CP-AMPA receptors can slow down considerably (3~4 folds) when assembled with γ -4 or γ -8 isoforms of TARPs (see Cho et al., 2007 *Neuron* 55:890; Milstein et al., 2007 *Neuron* 55: 905; Milstein et al., 2008 *Trends Pharmacol Sci* 29:333). Therefore, our data suggest that S-Type synapses in V1 L2/3 SOMs would likely represent CP-AMPA receptors co-assembled with either γ -4 or γ -8. This interpretation is consistent with the gene expression profile (Allen Institute for Brain Science's database: Transcriptomic cell types in the mouse brain: SMART-seq cells, <https://assets.nemoarchive.org/dat-8zpo593>) that show at least γ -4 is significantly expressed in a sizeable fraction of SOMs, which differs from PVs or L2/3 PCs. We now added this discussion to our revised manuscript.

3. The method of visual deprivation used in the study involved placing adult mice in the dark for seven days. Is it an appropriate method for inducing visual deprivation and whether it causes behavioral alteration? Normally, visual deprivation is performed during the critical period (P20- P35) by using monocular deprivation. What plasticity would happen if using the canonical method?

We had previously shown that a week of visual deprivation (dark-exposure, DE), is quite effective at producing large-scale cortical plasticity (including V1 L2/3) in adult mice (see Goel et al., 2006 *Nature Neurosci* 9: 1001; Goel & Lee, 2007 *J Neurosci* 27: 6692; Petrus et al. 2014 *Neuron* 81: 664; Petrus et al., 2015 *J Neurosci* 35: 8790), which we interpret as a model to support cross-modal plasticity (see Lee & Whitt 2015 *Curr Opin Neurobiol* 35: 119; Ewall et al., 2021 *Front Neural Circuits* 15: 665009; Lee 2023 *Front Synaptic Neurosci* 14: 1087042). Others have found that a similar duration of DE reinstates ocular dominance plasticity in adult rats and mice (see He et al., 2006 *J Neurosci* 26: 2951; He et al., 2007 *Nature Neurosci* 10: 1134; Huang et al., 2010 *J Neurosci* 30: 16636; Montey & Quinlan 2011 *Nature Commun* 2: 317). Therefore, DE is a robust model to study adult V1 plasticity. In contrast, as the reviewer mentioned, monocular deprivation (MD) is only effective at driving V1 plasticity during the critical period (P20-P35). The reason why DE is effective at enabling plasticity in the adult V1 is likely because it causes metaplasticity (see our reviews on this topic: Lee & Kirkwood 2019 *Front Cellular Neurosci*; Lee 2023 *Front Synaptic Neurosci* 14: 1087042). The plasticity of excitatory synapses on SOM by DE would suggest that it may play a role in the metaplasticity process that enables adaptation of V1 to visual deprivation. We now clarify these points in our revised manuscript.

4. Asynchronous release (AR) events generally increase with stimulus number due to residual calcium accumulation, making AR events rare after the initial stimulus. However, the initial AR frequency is around 5 Hz for all types of PC-SOM pairs in Fig. 4c. Is this due to the count of baseline EPSC events? If so, the baseline frequency should be removed.

We agree with the reviewer. We now reanalyzed the asynchronous release after removing the baseline spontaneous EPSC events, which is shown as Fig. 5 in the revised manuscript.

5. What is the role of different types of EPSC in shaping the action potential generated in SOM+ interneurons? Such as the firing probability or precision. The circuit function of different types EPSCs remains unknown.

To address this, we added a set of experiments where we played back uEPSC trains recorded from S-Type, F-Type, and Mixed PC to SOM connections to control SOM neurons and recorded their firing properties under current clamp. This shows that different types of uEPSC trains drive differential firing of SOM neurons dependent on the frequency of uEPSCs. As predicted from our short-term dynamic changes measured in voltage-clamp, we found that Mixed uEPSC trains produced the greatest number of action potential firing, followed by S-Type, and the least by F-Type uEPSC trains. As expected from their frequency-dependent facilitation, higher frequency trains generated more action potentials in SOMs. Furthermore, we found that DE causes a reduction in SOM firing to most of the uEPSC trains, except for F-Type EPSCs. This suggests that visual deprivation alters the functionality of SOMs dependent on the type of synaptic connections it receives from local PCs. The results are shown in new Fig. 4 and new Fig. S8. The results of statistical comparisons are presented in new Table S3 and pair-wise

comparisons across various condition/stimulation are presented in new Tables S4 and S5. These data further support the notion that visual deprivation causes differential adaptation of specific connection types on SOMs.

6. The original source of different types of EPSCs remains unclear. In addition to input differences, could subset differences of postsynaptic SOM+ cells contribute to distinct EPSC kinetics?

While we cannot completely rule out these possibilities, we believe these are rather unlikely based on several observations: (1) By using the SOM-GFP (GIN) transgenic line, we are confining our analysis to a subset of SOMs, which per Allen Institute for Brain Science's MET classification falls into a single category (Sst-MET-3). This suggests that there are no morphological, electrophysiological, and/or transcriptomic distinction among them that would warrant separation into different subtypes. (2) We clearly observe both S- and F-Type synapses within a single SOM cell with the proportion of these two types of synapses being observed across different SOMs in what seems to be a continuum (see Fig. 1g). Therefore, it is not possible to subdivide V1 L2/3 SOMs into subtypes based on the proportion of S- and F-Type synapses. (3) The majority of unitary connections from a single local PC to a single postsynaptic SOM contain both S- and F-Type synapses (categorized as Mixed connections). This further suggests that different synapse types are not segregated to distinct inputs. Whether there are inputs that strictly utilize S- or F-Type synapses will require a large-scale detailed connectivity mapping of various inputs to SOMs. (4) Furthermore, as we report here, the proportion of S/F-Type synapses undergoes plasticity with sensory experience (see Fig. 6c). Hence, we believe the varied proportion of these synapses on individual SOMs is driven by experience-dependent plasticity.

7. What is the relationship between baseline frequency and types of mEPSCs? For fast-type synapses, whether they hold higher baseline frequency thus causing uEPSC could not be distinguished from mEPSC. This could lead to the finding that the synaptic strength of F-type uEPSCs was only sensitive to stimulus number but not stimulus frequency. Because longer stimulation times involved more mEPSCs, ultimately leading to an increase in synaptic strength.

While this is a possibility, because mEPSCs are spontaneous events, F-Type mEPSCs would "contaminate" our measure of uEPSCs across all connection types. As shown in Fig. 6a, the frequency of F-Type mEPSCs are actually much lower than that of S-Types, hence it is unlikely that they would contribute much to the measurement of uEPSCs. Even when combined, the average frequency of mEPSCs in SOMs is 2~3 Hz (see Fig. S4b), which is low compared to that in other cell types. For example, in adult V1 L2/3 PCs average frequency of mEPSCs is ~4 Hz (see Petrus et al., 2015 *J Neurosci* 35: 8790) and in adult V1 L2/3 PVs it is ~20 Hz (Grier & Lee, *unpublished data*). Additionally, mEPSCs occurring at 2~3 Hz are highly unlikely to fall within one of the ten 3 ms periods following presynaptic action potential peaks, in which we measure our

evoked uEPSCs. Therefore, we believe our measures of uEPSCs are not likely affected by spontaneous mEPSCs.

8. Please give the full name when an abbreviation was used for the first time in the manuscript. (e.g., CP-AMPA)

We made the necessary edits.

9. Please ensure that all figure panels and corresponding legends are thoroughly reviewed to confirm their accuracy. (e.g., Fig. S6b)

We have corrected the figure and panel references that were mismatched in the text.

Reviewer #3

Major comment

1. The authors provide compelling evidence for the existence of F and S excitatory synapses on SOM inhibitory neurons. However, it remains unclear whether this phenomenon is unique to SOM cells or if it can be observed in other neuronal types, such as pyramidal neurons or parvalbumin (PV) positive inhibitory neurons, as well. Existing literature suggests that pyramidal neurons may exhibit similar categories of excitatory synapses as SOMs. To investigate this further, the authors could examine spontaneous mEPSCs in pyramidal and PV neurons in layer 2/3 of V1, applying the same analysis as performed for SOM neurons in Figure 1. These results could offer preliminary insights into the presence or absence of analogous excitatory synapse populations in different cell types and help determine whether this finding is a general feature of excitatory synapses or unique to SOMs in layer 2/3 of V1.

We apologize that this information was not clearly presented in our original manuscript. We had analyzed the mEPSCs from PCs and PVs using the same methodology as used for clustering mEPSCs from SOMs. This was presented in one of the supplementary figures (now Fig. S10a). As presented in Fig. S10, we compared this to the analysis of mEPSCs from SOMs (now Fig. S10b). As shown in Fig. S10d, mEPSCs from PCs look similar to S-Type synapses in SOM, while those from PVs resemble F-Type synapses on SOM.

We have now run an additional analysis of the mEPSCs from PCs and PVs after pooling them together and clustering them into two groups using the same method used on SOM mEPSCs. The results are shown in the new Fig. S10c. The difference is negligible when plotted in the same manner as for SOMs (please compare to Fig. S10a), and the error for misclassification is

restricted to the lower left corner of the graph, where the PC mEPSCs are in comparatively higher proportion as compared to SOM and PV mEPSCs.

Minor comments

2. In the introduction, the authors highlight the "critical role of SOM" (and VIP) inhibitory neurons in cross-modal sound-mediated receptive field sharpening in V1, citing Ibrahim et al., 2017 as evidence. However, the cited study does not establish a crucial role for SOM neurons in cross-modal multisensory processing, as it states that "neither PV nor SOM neurons appeared to contribute significantly to the sound-induced effect." Furthermore, the paper does not present data on cross-modal receptive field sharpening, but instead investigates the impact of sounds on orientation selectivity of individual neurons. Although the orientation selectivity of a neuron is closely related to its receptive field, the authors should (in my opinion) interpret this finding more carefully. As such, the authors should consider rephrasing this statement or emphasizing the potential importance of SOM neurons in multisensory processing from a different perspective.

We apologize for over-interpreting the previous finding. We have removed the statement.

3. It would be beneficial to depict an image of a brain slice containing V1, in which MCs expressing GFP are visible, in either Figure 1 or Supplementary Figure 1. This addition would make the related Figure more intuitive, allowing readers to immediately identify the cell type of interest in this study.

We have added an image of a V1 slice and 3D reconstruction of some of the V1 L2/3 SOM-GFP neurons (see new Fig. 1a).

4. In Figure S4b, the authors show that Naspam application substantially diminishes mEPSC amplitudes in SOM neurons on a global scale, which suggests the existence of CP-AMPA receptors in their membranes. To determine if Naspam preferentially affects mEPSC amplitudes of either S-type or F-type excitatory synapses, the authors could analyze the data separately for each synapse type. This distinction would offer preliminary evidence regarding whether CP-AMPA receptors are predominantly involved in either S-type or F-type responses.

We thank the reviewer for the suggestion. As mentioned in detail in our response to Reviewer 1 (comment #2), we reanalyzed the data by clustering the mEPSCs and found that S-type synapses are selectively reduced by the CP-AMPA blocker. This suggests that SOMs contain rather unusual slow CP-AMPA receptors, which are likely co-assembled with the γ -4 or γ -8 isoforms of TARPs. Please see our response to Reviewer #1 for more details.

REVIEWERS' COMMENTS

Reviewer #1 (Remarks to the Author):

The authors have addressed my concerns and the additional new data makes the paper stronger.

Reviewer #2 (Remarks to the Author):

The authors have addressed all my concerns.

Reviewer #3 (Remarks to the Author):

The authors have adequately addressed all of my concerns. I have just one minor comment remaining.

Minor comment

2) In Fig. 1a's left panel, a confocal image of a V1 section counterstained with DAPI (blue) is presented. The delineation of the six cortical layers, as shown by the authors, appears to be incorrect. Typically, one would expect the highest density of DAPI-stained nuclei in layer 4. However, in the presented figure, layer 4 exhibits a notably low density, aligning more with layer 5. Consequently, the designated upper and lower boundaries of layer 4 should be adjusted upward, to align more with the bottom portion of what is currently labeled as layer 2/3, where the density of DAPI-stained nuclei peaks.

Moreover, the related figure caption states that GFP-positive neurons are primarily observable in layer 2/3. This description will be inaccurate once the layer boundaries are adjusted. In the corrected version, the majority of GFP-positive neurons would be situated in layer 4. This discrepancy requires rectification.

Response to Reviewers' Comments

We are pleased that the reviewers found that we had addressed all of their concerns. Reviewer #3 found the border assignment of L4 was not done correctly, which we have addressed as detailed below.

Reviewer #1 (Remarks to the Author):

The authors have addressed my concerns and the additional new data makes the paper stronger.

We thank the reviewer to stating that the additional data strengthened the paper.

Reviewer #2 (Remarks to the Author):

The authors have addressed all my concerns.

We thank the reviewer.

Reviewer #3 (Remarks to the Author):

The authors have adequately addressed all of my concerns. I have just one minor comment remaining.

Minor comment

2) In Fig. 1a's left panel, a confocal image of a V1 section counterstained with DAPI (blue) is presented. The delineation of the six cortical layers, as shown by the authors, appears to be incorrect. Typically, one would expect the highest density of DAPI-stained nuclei in layer 4. However, in the presented figure, layer 4 exhibits a notably low density, aligning more with layer 5. Consequently, the designated upper and lower boundaries of layer 4 should be adjusted upward, to align more with the bottom portion of what is currently labeled as layer 2/3, where the density of DAPI-stained nuclei peaks.

Moreover, the related figure caption states that GFP-positive neurons are primarily observable in layer 2/3. This description will be inaccurate once the layer boundaries are adjusted. In the corrected version, the majority of GFP-positive neurons would be situated in layer 4. This discrepancy requires rectification.

We thank the reviewer for pointing this out. As reported in previous studies (e.g. Ma et al., 2006 J Neurosci 26:5069-5082), the GIN line expresses GFP in SOM neurons across different layers including L2/3. The initial image we had included in the figure made the new border assignment appear too superficial (we found that the image was closer to the medial border of V1 where L4 tapers up). We now replaced the figure with another one that is away from the medial border and realigned the lower L4 border where the density of DAPI starts to decrease (~500 μm depth from the pia) as recommended by the reviewer. We also modified the Fig. 1a legend to remove the statement.